# Research in Computational Expressive Music Performance and Popular Music Production: A Potential Field of Application?

Pierluigi Bontempi, Sergio Canazza *, Filippo Carnovalini and Antonio Rodà

Department of Information Engineering, University of Padua, 35122 Padova, Italy
* Correspondence: canazza@dei.unipd.it

**Abstract:** In music, the interpreter manipulates the performance parameters in order to offer a sonic rendition of the piece that is capable of conveying specific expressive intentions. Since the 1980s, there has been growing interest in expressive music performance (EMP) and its computational modeling. This research field has two fundamental objectives: understanding the phenomenon of human musical interpretation and the automatic generation of expressive performances. Rule-based, statistical, machine, and deep learning approaches have been proposed, most of them devoted to the classical repertoire, in particular to piano pieces. On the contrary, we introduce the role of expressive performance within popular music and the contemporary ecology of pop music production based on the use of digital audio workstations (DAWs) and virtual instruments. After an analysis of the tools related to expressiveness commonly available to modern producers, we propose a detailed survey of research into the computational EMP field, highlighting the potential and limits of what is present in the literature with respect to the context of popular music, which by its nature cannot be completely superimposed to the classical one. In the concluding discussion, we suggest possible lines of future research in the field of computational expressiveness applied to pop music.

**Keywords:** computational music expressive performance; popular music; music production; digital audio workstation; virtual instruments

## 1. Introduction

When a skilled musician plays a piece of music, s/he usually does not do so mechanically, keeping a perfectly steady timing without any concession to loudness, timbre variations, or (possibly) the use of embellishment techniques. Expert musicians commonly manipulate at least some performance parameters to produce expressive sonic renditions of the pieces:

*[...] performers are able to use systematic variations in performance parameters to convey emotion and structure to listeners in a musically sensitive manner* ([1] p. 64).

With regard to western classical music, traditionally expressiveness has been defined in terms of deviations from what is prescribed in the score:

*When playing a piece, expert performers shape various parameters (tempo, timing, dynamics, intonation, articulation, etc.) in ways that are not prescribed by the notated score, in this way producing an expressive rendition that brings out dramatic, affective, and emotional qualities that may engage and affect the listeners* ([2] p. 1).

On the other hand, in the context of popular music, the traditional separation between the composer, score author, and performer who interprets and gives a sonic rendition is only rarely fully appropriate. A situation partially comparable to that of the classical world occurred only before the diffusion of electric recording (between the end of 1920s and 1930s), when the popular music business focused on songs, sheet music sales, and related publishers, but not on recording artists or specific records [3].

Generally speaking, in pop music parts can be improvised, can be based on more or less detailed lead sheets (an essential form of musical notation that describes the fundamental elements of a song [4]), can be created and memorized by the musician without the need of a formal notation. Parts can also arise from experimentation in the studio and interactions between engineers, producers, and musicians [3]. Moreover, with the spread of computer-assisted music production, popular music creation has shifted toward a model in which the traditional composer/performer distinction further loses meaning, in favor of the deep integration of activities associated with music composition, engineering, production, and performance [5].

The very definition of the authorial role in pop music is more fluid and complex than solid and univocal [6].

In popular music, it is neither simple nor correct to separate the author and performer, a concept that many EMP literature studies are based on; however, this does not mean that performative expressiveness is not of primary importance in the former context. There are some specific exceptions: in electro-pop "deadpan" performances, commonly considered inexpressive and not desirable in classical music, can fall within sought-after aesthetic intentions, as well as the construction of musical parts that can only be played by machines and not by humans [7]. Apart from these specific cases, human-like expressivity in popular music performances plays a crucial role.

Within computer-assisted music production, the producer often creates (through mouse and keyboard editing and/or use of dedicated hardware controllers) MIDI parts associated with virtual instruments [8].

Simply using the mouse or the keyboard to insert into the rhythm grid of the digital audio workstation (DAW) the MIDI notes that the virtual instrument will have to sonorize is not enough to produce something that can be perceived as expressive. At the same time, it is not easy to obtain realistic and expressive-sounding parts using a MIDI controller because of the possible technical limitations of the controller and/or the skills required by the performer. Manually refining parts so that they are more expressive is only partially possible, and it is undoubtedly tiring and time-consuming.

Commercially available tools (ones commonly used by pop producers, see Section 2 for some examples) only partially cover the needs related to the creation of expressive performances, which do not sound robotic or too flat and uninteresting, are engaging, and integrate better with human played parts while also respecting the emotional or more generically expressive intention of the piece or of a specific passage.

Innovative solutions capable of automatically or semi-automatically conferring expressiveness to the MIDI parts produced would be highly valuable, and it is our opinion that existing academic research on expressive music performance (EMP) could help improve the tools available or suggest directions to develop new ones.

The following part of this contribution is organized as follows. In Section 2, an analysis of the strengths and limitations of what is commercially available in relation to musical expressiveness in computer-assisted popular production is offered. In Section 3, existing EMP literature is reviewed; the contributions are organized in thematic subsections, such as the relation between expressiveness and structure, local expressiveness, the relationship between emotional intention and expressive parameters, and so on. At the end of each subsection, a comment about the possible applications or limits of the EMP literature presented in relation to popular music computer-assisted production is provided. Section 4 presents the general discussion and conclusions.

## 2. Commercial Products

Since the early 2000s, digital audio workstations have become central to popular music production workflows, and are now fundamental working tools for professionals as well as "bedroom" producers [9]. Among the benefits of DAWs, there is the ability to work with a wide variety of realistic-sounding virtual instruments, which can be used for creating complex arrangements [10].

Virtual instruments can be based on audio samples of real instruments as well as on sound synthesis techniques [11]. The virtual studio technology (VST) standard is probably the most widespread and is used to build virtual instruments and audio effects [12].

When a popular music producer uses a virtual instrument in an arrangement, s/he can program the MIDI part asynchronously, or play the part in real-time using a MIDI controller (usually, but not necessarily, a keyboard-shaped one [13]). In both cases, making the part sound expressive is not a trivial task, despite the extensive tonal and dynamic potential of modern virtual instruments.

To obtain an updated list of tools and practices commonly used in popular music production and concerning expressiveness, we entered the following keywords in the search fields of the websites of some of the most accredited sector magazines: Sound on Sound (https://www.soundonsound.com, accessed on 3 October 2022), Tape OPpathogens-2129028 (https://tapeop.com/, accessed on 3 October 2022), MusicRadar (https://www.musicradar.com/, accessed on 3 October 2022), which includes the magazines Future Music, Computer Music, and Music Tech:

- Expressiveness;
- Expressive performance;
- Virtual performance;
- Virtual performer.

Up to sixty results were consulted (when available) per search key per portal. Once the results that were not relevant to this contribution were filtered, examples that were useful for providing a general overview of the sector were selected from the rest. In the event of the presence of distinct commercial products, but similar in terms of operating principles and construction/development logic, not all of them have necessarily been mentioned, as the objective of this contribution is not to offer a complete overview of what the market offers, including each individual product, but to account for the main tools associated with expressiveness generically available to contemporary music producers. The results of the research are categorized in thematic areas and are presented in the following subsections of Section 2.

### 2.1. Common Tools

With recent virtual instruments, some basic parameters can be controlled using MIDI messages and/or DAW automation, namely equalization (EQ) and a high-pass filter (HPF), possibly with settable resonance (and, sometimes, compression). More advanced sound control potential may be present.

It is common for sampled instruments to have multiple samples in each note, usually selected and loaded based on the MIDI velocity of the note played. Sample-based virtual instruments frequently implement the round-robin technique (when multiple consecutive requests of the same note are received, the instrument loads different samples, in random or predetermined order, to avoid the so-called "shotgun effect", the repetition of a series of identical sounds, an event perceived as unnatural in the musical field).

DAWs can usually randomize MIDI data, in compliance with the range of relative values set by the user. This is sometimes done to alleviate the feeling of unnaturalness that quantized, fixed-velocity parts transmit [14].

All of these expressive tools and tricks need to be triggered (in a static or—when possible and appropriate—dynamic fashion) using offline editing or MIDI controllers [15]. That translates into a wide range of possible sound nuances, which must be controlled manually by the producer, be it in real time or not.

### 2.2. Triggerable Instrument-Specific Patterns

Especially among virtual instruments that model physical ones, the presence of patterns that can be recalled by the user is frequent. These are normally pre-recorded or pre-programmed performances of short parts, usually structured so that they can be repeated in a seamless loop. Sometimes, but not necessarily, these patterns can be customized

by the user, directly within the virtual instrument or using the MIDI editing tools of the DAW.

Some of the products that include this functionality are:

- Steinberg Groove Agent 5;
- UJAM collection (Virtual Guitarist, Virtual Bassist, Virtual Drummer, Virtual Pianist);
- Native Instruments Action Strings and Emotive Strings.

This approach, which could be considered an evolution of traditional arpeggiators, can be useful among producers that are not able or interested in writing the instrument parts note-by-note. Moreover, some degree of expressivity can be included in the patterns—what kind exactly depends on how the pattern is realized and on the sonic capabilities of the virtual instrument. Sometimes the patterns can be modified in real-time setting parameters such as complexity and intensity in Steinberg's percussive-oriented sampler Groove Agent 5 (Figure 1): Complexity (x-axis) loads richer, more nuanced patterns as you move to the right, while Intensity (y-axis) affects the MIDI velocity of the strokes.

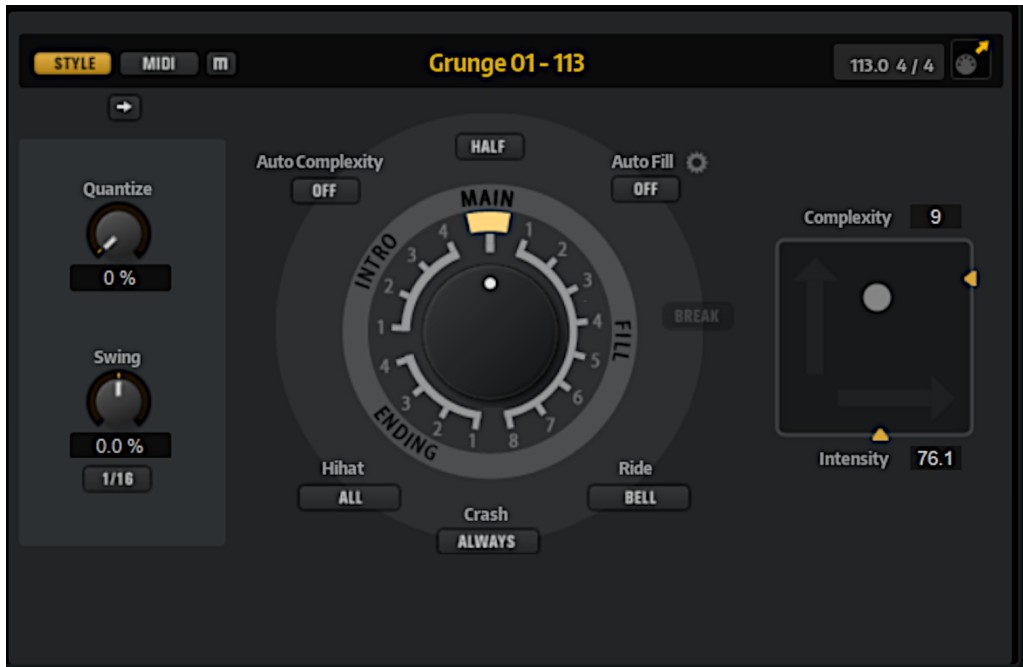

**Figure 1.** The pattern management area in Steinberg Groove Agent 5. On the right, the two-dimensional control surface (that also works in real time) is dedicated to the intensity and complexity parameters. Courtesy of Steinberg Media Technologies GmbH.

The review of the UJAM Virtual Bassist collection published on Sound on Sound ([16] p. 130) says:

> *[. . . ] my completely virtual band of session musicians just needing pointing in the right direction. Add in some 'human' with a few guitar overdubs and some vocals, and a song idea can be fleshed out very quickly. Moreover, the virtual band sounds very polished indeed.*

This quotation highlights two common limitations of the pre-made pattern approach. It can hardly be completely "tuned" to the expressive intention of the specific piece of music under processing (making a human contribution necessary, in the form of human-played added parts or manual editing of the patterns), and it usually sounds very polished, maybe even too polished to be perceived as truly "real". Moreover, having pre-established patterns, albeit sometimes customizable, places this type of resource between expressiveness and automatic/assisted music generation [17], touching territories outside of the boundaries of this contribution.

### 2.3. Triggerable Instrument Articulations

In the jargon of contemporary music production, *articulations* mean different timbres or performative techniques that can be associated with the virtual instrument. For example, the same notes can be played by a virtual violin (among other possibilities) with *detaché*, *staccato*, or *col legno* articulations.

This approach is very common among sample-based virtual instruments belonging to the classical orchestra category (strings, brasses, woodwinds), but can potentially be found in about any kind of virtual instrument.

Commercial products falling into this category are offered by manufacturers such as EastWest, IK Multimedia, Vienna Symphonic Library, Steinberg, Native Instruments, and Spitfire Audio. Similar to what has been said in the 'Common Tools' section 2.1, also in articulations triggering a manual intervention of the producer is almost always needed.

One exception is that of the Synful Orchestra, whose operating principle has been described in detail by the creator E. Lindemann [18]. The sounds are not given here by PCM audio samplings (except for the noise part), but are obtained by additive synthesis, on the basis of harmonic amplitude envelopes extracted from recordings of real instruments. Slowly varying pitch and amplitude controls are derived directly from MIDI input, and converted into sound using additive synthesis. On top of that, to take care of the rapid fluctuations in loudness, pitch and timbre that usually occur during the transitions from one note to the next one, the Reconstructive Phrase Modeling (RPM) is proposed. It searches a database of instrumental phrases, finding fragments similar to what is requested via MIDI, modifying them appropriately and combining them together. In this way, according to the author, the main problems of the sampled instruments are solved, i.e., long notes that vary over time, a passage from one note to another that is fluid and natural, and not the juxtaposition of separate sound events. The Synful Orchestra is a rare example of attention to the expressive dimension which is placed—at least partially—at the level of synthesis or sound generation and not at that of the control of the virtual instrument. Indeed, the orchestra can actually play notes with different articulations, but operate autonomously on the basis of the incoming MIDI notes, without the need for explicit triggers.

### 2.4. Advanced Hardware Timbre and Expressivity Control

On the controller front, many attempts to produce innovative and powerful MIDI devices, sometimes called hyper-instruments (term used in particular in the case of traditional instruments modified to act as MIDI controllers [19]), capable of manipulating multiple parameters in real time, can be cited:

- The pioneering Kurzweil XM1 Expression Mate, which dates to 2000;
- ROLI Seaboard family, whose operation is traditionally based on the MPE (MIDI Polyphonic Expression) protocol (Figure 2);
- Expressive E Touché and Osmose;
- Keith McMillen SoftStep 2;
- Erae Touch (which makes use of MIDI 2.0).

These tools can be touch and/or velocity-sensitive, and often allow the control of pitch bend, polyphonic aftertouch, microtonal slides, and in general, MIDI control changes/continuous controllers (CCs) through gestures.

In academia, which can have (and have had) a direct impact on commercial products, the International Conference on New Interfaces for Musical Expression (NIME—https://www.nime.org/, accessed on 19 December 2022) (held since 2001) is of particular relevance. It has become the point of reference for researchers dedicated to the interface design and human–computer interactions in music production [20]. A clear example of the possibility of this field of scientific research in having a direct effect on the commercial world is that of Roli, Ltd.: the initial project of the Seaboard keyboard, which would later introduce a disruptive element of novelty in the production of pop music, was first presented at NIME [21].

Once again, these tools are designed to give the producer direct control of the parameters of the virtual instruments; they are neither able to generate autonomous musical expressiveness, nor are they usually aimed at that.

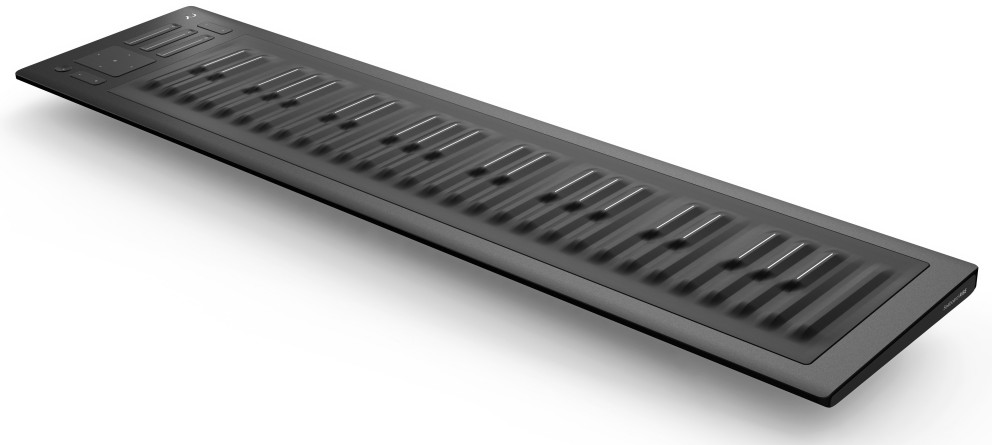

**Figure 2.** The MIDI controller ROLI Seaboard Rise. Courtesy of ROLI, Ltd.

### 2.5. Automatic Analysis of the Rhythmic and Harmonic Structures and Generation of New Musical Parts

In some cases, not particularly widespread, the virtual instrument is able to automatically generate parts on the basis of the data obtained from the analysis of the parts of other instruments or harmonic progression.

One of the best representatives of this approach is Toontrack EZ Bass (Figure 3), which can analyze MIDI or audio parts and generate matched bass lines, and is highly expressive thanks to the use of different articulations, embellishments, velocity variations, etc. Similar to what has been said about pattern-based virtual instruments, in this case, it is questionable whether this kind of capability should be included in the automatic composition/arrangement area, or the expressive performance one. Nonetheless the bass lines generated by the software present expressiveness, and – even more important – are context-aware, at least with regard to some parameters/musical dimensions, things that are of prime relevance in the field under consideration. Indeed, in popular music, instruments and vocals are rarely isolated. In the vast majority of cases, they are included in arrangements that involve the overlapping of many distinct but synergistic elements. The interactions between different instruments will be addressed in relation to scientific research in Section 3.10.

### 2.6. The Missing Link

From what has been observed in the previous subsections it is evident that, in the current panorama, extremely sonically versatile virtual instruments with rich expressive potential are available on a commercial level, which must be controlled in real time (there is no shortage of hardware tools for sophisticated real-time control) or through offline MIDI sequencing by the producer. Many of them provide pre-established—sometimes customizable—patterns, and some can generate context-aware ones. However, these last features, more than in EMP, fall within an area between expressiveness and automatic or computer-aided composition. What is missing here are tools capable of processing musical parts provided by the producer—potentially time-quantized and fixed-velocity ones—making them expressive, acting automatically on the tone parameters of the virtual instrument, time deviations, and (possibly) the automatic selection of appropriate articulations or embellishments. To be completely effective, these potentials should then be sensitive to the context, and possibly to high-level indications (for example, to emotional intentions) provided by the producer. Could academic research in computational expressive music performance help? Considering that, historically, EMP has mostly addressed

classical music for solo instruments, which research lines could find direct application in popular music? Which adaptations, if any, would be needed?

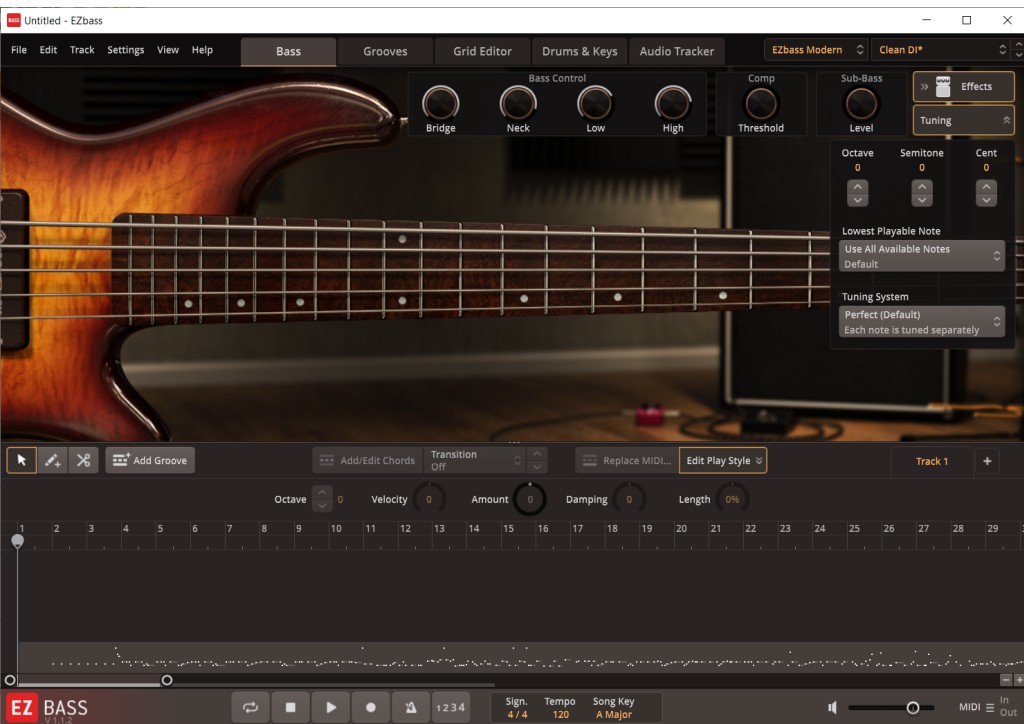

**Figure 3.** Toontrack EZ Bass. Courtesy of Toontrack Music AB.

## 3. Research Products

### 3.1. A Multifaceted Field of Research

Computational expressive music performance (EMP) involves different approaches and areas of expertise. Several academic disciplines are involved (computer science, musicology, psychology). It is, therefore, not surprising that academic researchers have generated diverse approaches with respect to the type(s) of technology employed and the specific aspect(s) of computational EMP addressed.

The first scientific contributions to the topic date back to at least the 1980s and more than one review work has already been published [2,22–26]. There are two main purposes associated with EMP computational models. They can be used as analytical tools for understanding how humans perform music or generate new performances of musical pieces (in many different contexts) [2]. The two are connected, i.e., in developing models capable of credible virtual performances, it is necessary to understand what makes human performances expressive and worthy of interest. Moreover, concerning the technologies involved in the existing models we can identify, generally speaking, two areas: data-driven and rule-based models [2]. The former category relies on machine learning, probabilistic, and artificial neural network (ANN) approaches (and, consequently, large collections of data), the latter on manually designed rules, based on musical hypotheses. Another commonly used nomenclature contrasts the *analysis by synthesis* to the *analysis by measurement*. The former indicates the implementation of rules obtained from dialogue with human experts, the latter indicates the use of real performance parameter measurements to extract rules or other significant regularities.

Below, the main research lines that have been followed by scholars will be presented at a medium- to high-level of abstraction. They will be divided into the following thematic areas:

- Visual representation of expressive performance features;
- Relation between expressiveness and structure of the musical piece;
- Local expressiveness;

- Score marking interpretation;
- Relationship between emotional intention and expressive parameters;
- Relationship between sensory experiences and expressive parameters;
- Identification and modeling of the performative styles of real musicians;
- Identification of physical and psychological limits of the performer;
- Ensemble music modeling;
- Conductor systems.

We believe that this subdivision (i.e., into thematic macro areas) could be functional in providing a clear understanding of the computational EMP field, despite the fact that the musical parameters investigated may be common in all areas (timing, loudness, timbre, etc.). What changes is the logic that leads an approach to study or modify the specific parameters. Moreover, some of the research lines reviewed here address more than one area at the same time and can, therefore, appear several times during the course of treatment, in different subsections.

The main computational EMP systems under examination will then be recalled in Section 3.12, where for each of them a summary of the main characteristics will be provided in Table 1, specifically with regard to:

- Technologies involved;
- User interaction;
- Main goal(s).

A color-based association between the systems in relation to the thematic macro areas will also be included.

### 3.2. Visual Representation of Expressive Performance Features

Although a visual representation of expressive performance features is not a computational model for expressiveness, its relevance in the understanding of human-performing dynamics and the subsequent construction of models must not be underestimated. Graphical representations of expressive parameters could be relevant to the objectives of this contribution, not only because they offer an intuitive and immediate tool for understanding expression dynamics that may be hard to catch otherwise by a human observer, but also because they could simplify the intuition and understanding of the analogies and differences between musical expressiveness in the historical/stylistic/instrumental fields to which academic research has traditionally turned, and what happens in a popular context. This is particularly true for solutions that offer graphical representations of more than one expressive parameter at a time since the overall perception of the performance is linked to the interactions between several parameters rather than to decontextualized–individual ones [27].

A system for the real-time representation of performances in a two-dimensional graph (with the tempo (bpm) on the x-axis and loudness on the y-axis) was proposed by J. Langner and W. Goebl [27]. Along the two axes, a dot moves in synchrony with sound. The materials analyzed were piano performances played on a Bösendorfer computer-controlled grand piano (SE290). Timing information was extracted from MIDI data, while loudness data were extracted from the audio files of the performances. The trajectory of the dot (that gradually fades away, leaving behind a visible tail) is a visual description of the two most important parameters of the performance, tempo, and loudness.

This kind of visual representation has been subsequently taken up and evolved, with the extraction of time information directly from audio, and not from MIDI data, and the coinage of the term *performance worm* [28], which has also been used in other scientific publications [29] (see Figure 4).

Non-standard music transcription techniques, indicated for repertoires handed down orally or to report dimensions that escape standard Western notations, are of primary interest in the field of ethnomusicology [30]. With necessary differences (symbolic representation instead of a direct *analogical* representation of the phenomenon [31]), there is the question of moving performative sound dimensions into the visual sphere. It does

not seem unreasonable for us to think that reflections of this kind could have positive repercussions in the EMP field, although the goals are clearly different in the two cases. One way of representing unambiguously the desired prosodic interpretation of melodies using a dedicated small alphabet A = {l⁻, l$^x$, l⁺, l$^→$, l$^←$, l$^*$} and a deterministic mapping from the prosodically labeled score to sound synthesisis has been proposed by Christopher Raphael [32] (see Figure 5).

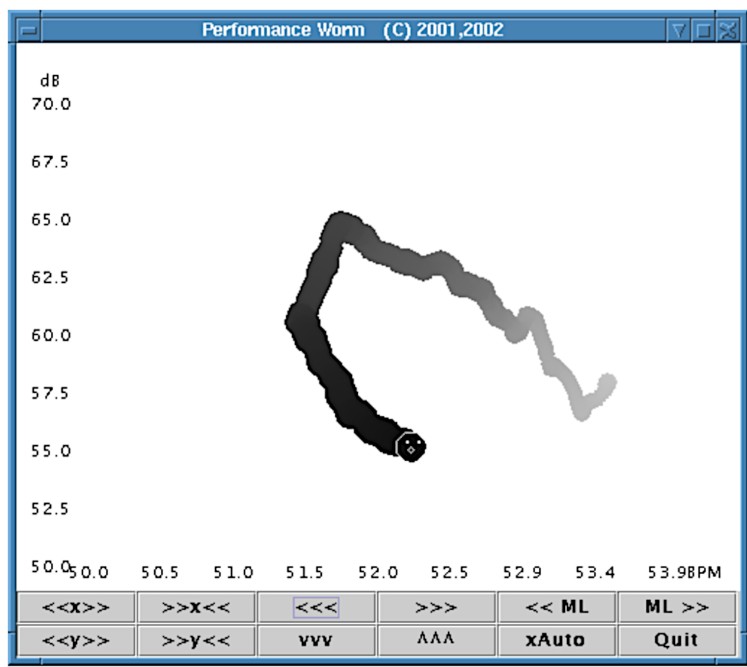

**Figure 4.** A screenshot of the performance worm. The x-axis shows the tempo in beats per minute; the y-axis shows the loudness in decibels [29].

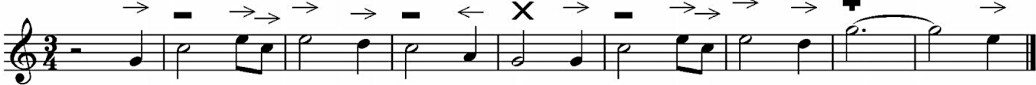

**Figure 5.** The first few measures of the melody of the popular tune Amazing Grace, showing a custom note-level prosodic labeling. Reprinted with permission from [32]. 2010, Taylor & Francis.

While integrated graphical representations of expressive parameters (such as the ones related to the *performance worm* [27–29]) could be relevant in the implementation of popular music expressive models for the reasons seen above (they simplify the intuition, understanding, and comparison of expressive phenomena), they can also find a more direct application in the making of conductor systems, as described in Section 3.11.

Moreover, the reflection on the notation of dimensions in the musical performance not included in Western standard writing can offer useful insight into the development of EMP models (particularly for conductor systems).

### 3.3. Relation between Expressiveness and Structure of the Musical Piece

Pieces of music can be described in terms of form or structure at different levels of abstraction. At the highest level, we have the sections of the piece, as in the classic sonata form, divided into introduction, exposition, development, recapitulation, and coda sections. At a lower level that is probably more relevant for computational music expression, we can identify structural elements such as, from the simplest to the most structured, motifs, phrases, and periods [33–35].

Quite often, the clarification of the piece structure is considered one of the main aims of expressive performance [36]. A radical (non-computational) approach, which substantially

brings the performative expressiveness back to the structure of the piece, is that of E.F. Clarke [37], who bases the discussion on generative principles [38], in turn influenced by the Schenkerian analysis.

Many attempts at modeling expressive parameters based on structural analysis are present in the literature. Tempo changes related to phrase structures in tonal music were at the center of the seminal research of N.P.M. Todd [39]. The author's approach is based on Lerdahl and Jackendoff's generative theory [38]. The piece is divided into nested hierarchically organized time spans. The model reflects the structure of the piece slowing at structural endings, in a more or less pronounced way depending on the hierarchical importance of the syntactic break (see Figure 6). Subsequently, Todd expanded his approach to include the computational modeling of rubato in relation to the music structure [40] and dynamics, based on the assumption that there is a correspondence between speed and intensity (the faster, the louder) [41]. Although subsequent publications have shown the relevant limitations of Todd's model [42,43], this remains a cardinal point in the evolution of the research field of computational musical expressiveness.

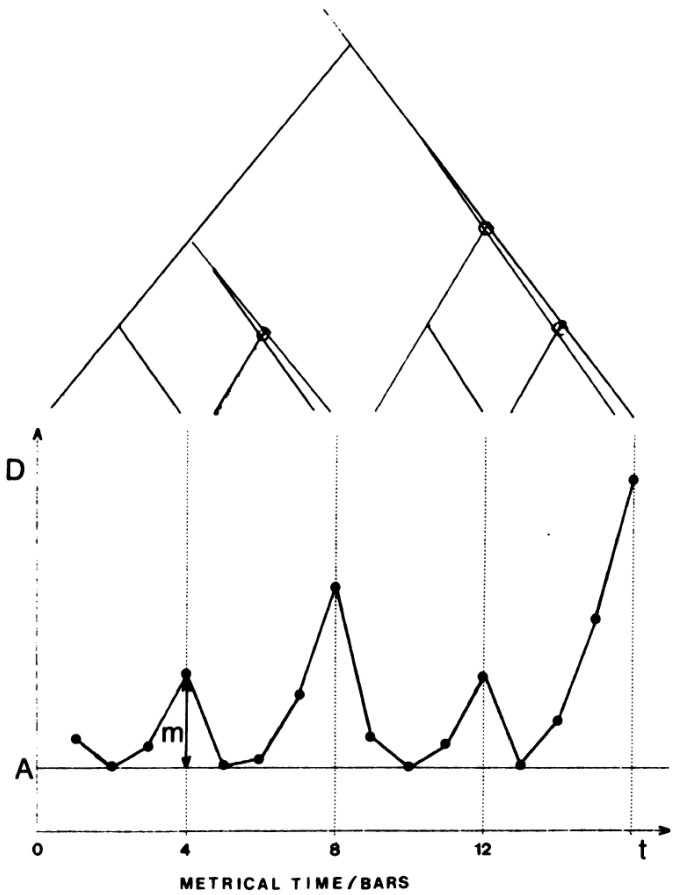

**Figure 6.** A tree diagram representing the hierarchical structure of the piece and, below, the corresponding tempo variations generated by the model (the higher the value on the y-axis, the more pronounced the slowdown). Reprinted with permission from [39]. 1985, University of California Press.

G. Grindlay and D. Helmbold proposed a hierarchical hidden Markov model (HHMM) to extract statistical data about the relations between the score structure and associated performances (in particular with regard to time deviations) [44]. The model is based on a two-level hierarchical structure, with the top level representing the musical phrase context, while the lower one is associated with note-level contexts. Once trained, the model is able to generate expressive performances, but can also be used to recognize individual performers.

A similar approach, based on the distinction and relation between a phrase level and a note level, can also be declined in a rule-based system, as shown by G. Widmer and A. Tobudic [45,46]. Note-level and phrase-level expressive patterns are combined to generate predictions about complex composite expression curves for new pieces.

One of the most relevant EMP lines of research is that of the KTH rule system for musical performances [47]. Substantially based on the *analysis-by-synthesis* approach [48] (simplifyable into the selection and analysis of the performances and the variables to study → tentative synthesis of varying versions through the definition of rules →, human judgment of the performance, and the iterative return to the previous phase to improve the model), the system also provides a complementary *analysis-by-measure* approach (the rules are based on objective data derived from real performances). The rules affect various parameters, such as timing, dynamics, and articulation, and they are weighted. They are organized in the following categories: phrasing, micro-level timing, metrical patterns and grooves, articulation, tonal tension, intonation, ensemble timing, and performance noise; therefore, they cover the piece of music at different levels. Specifically, in relation to the structural dimensions, the KTH system suggests rules for the creation of phrase arch-like tempo and loudness contours, and for the insertion of a *ritardando* at the end of the piece.

F. Carnovalini and A. Rodà proposed a system capable of generating brief melodies together with their expressive performances, all based on a multilayered hierarchical subdivision of the notes reminiscent of the Schenkerian approach [49]. The length and loudness of the notes are automatically set according to their hierarchical relevance, within a three-level model. The KTH phrase-arc rule is also implemented.

The structure of a piece can often be analyzed in terms of the construction and resolution of tensions (a task usually easier within tonal music, associated with a more general logic of construction and resolution of expectations [50,51]). Computational expression models can refer to those tension patterns.

A three-dimensional spiral representation of pitch classes, chords, and keys to compute variations in tension during the piece of music was introduced by D. Herremans and E. Chew [52]; based on that, a computational study of the role of tonal tension in the prediction of expressive performances of classical piano music was presented by C. Cancino-Chacón and M. Grachten [53]. A computational model for the calculation of tempo and dynamic variations was implemented using a bidirectional LSTM recurrent neural network (which processes both backward and forward data). The use of tonal tension, as defined by Herremas and Chew, is useful for predicting expressive changes in tempo and dynamics, but not to predict specific values for those parameters. Tonal tension seems to be an additional—but not self-sufficient—resource in the creation of models for musical expressiveness.

The relation between music performance expressiveness and piece structure could be as relevant in popular music as it is in classical music, but the subject needs deep analysis and investigation before being able to draw any conclusions since the two contexts are only partially superimposable from a structural perspective. Moreover, the tonality-related dynamics enhanced in classical music by some researchers could be relevant only in specific popular contexts, but would probably be out of focus and of little help in many others [54]. For this reason, the applicability of computational solutions for the automatic identification of the structure of the piece (aimed at the subsequent use of expressive models) based on tonal music theories [55,56], may not be ideal in many cases in popular music. The melodic analysis model suggested by Carnovalini and Rodà [49], based on the previous works of theirs and N. Orio and F. Simonetta [57,58], seems to be much more promising with regard to popular music, since it represents a more theory-agnostic approach. It classifies the notes of the piece within a three-level hierarchical structure based on the metric position, relevance of the underlying chord with regard to tonality (but this could be easily adapted to the modern modal approach to popular music composition), and the relevance of the melodic note within the underlying chord.

*3.4. Local Expressiveness*

Musical expressiveness can also be observed and modeled locally, note-wise, or in relation to small groupings of notes.

Fundamental in this perspective were the seminal studies of A. Gabrielsson - among many others [48,59–61]. Gabrielsson's approach is measure-based: an analysis of real performances, looking for relevant performance variables makes it possible to detect systematic variations (SYVARs) related to some type of norm. These variations (or deviations from mechanical regularity) may vary according to the specific context (type of music, performer, etc.). Gabrielsson investigated rhythmic micro-structures (e.g., a half note followed by a quarter one, or a dotted chrome followed by a sixteenth note, etc.), as well as deviations at half- and full-measure levels, finding significant regularities, at least given the same context. Gabrielsson observed that:

> *It seems safe to assume that such differences in performance are not primarily made in order to affect the perceived structure, but rather to contribute to a proper motional-emotional character of the music in question. ([48] p. 79)*

Generating systematically varying sound sequences based on the findings of the analysis and subjecting them to human evaluation allows testing of the validity of the detected SYVARs and the relationships between them and the experiential psychological variables associated with the listener.

The expert system approach has also been investigated. In M.L. Johnson's model for the expressive rendition of Bach's fugues [62], the rules, based on the expertise of two professional performers, affect tempo and articulation, and are associated with specific rhythmic patterns.

Many of the rules of the KTH system [47] also address the *local EMP area*. For example, the duration contrast rule does "*Shorten relatively short notes and lengthen relatively long notes*", the Double duration one does " *Decrease duration ratio for two notes with a nominal value of 2:1*". A combination of the rule-based approach with artificial neural networks and user interaction has also been tried [63].

It seems that structure-based expressiveness (see previous section) can be more easily appreciated by following an analysis by synthesis approach, while local expressiveness is more prone to data-driven approaches, at least taking into consideration current technologies and the studies already carried out. This is confirmed in work by S. Oore et al. [64], where an LSTM network used to generate both piano solo musical parts (automatic composition) and their expressive performances showed better results on a local basis, more than in a long-term structure. In a proposal by K. Teramura et al. [65], a machine learning Gaussian process regression predominantly based on local input data (durations and pitches of the notes and of those belonging to the previous and subsequent measures were analyzed, as well as more general indicators, such as the meter and melodic line) was used to render expressive performances of music scores. Similar parameters were considered in the statistical model YQX [66], together with principles taken from E. Narmour's implication-realization model [51]. Binary tree-based clustering was explored by K. Okumura et al. to classify the local context of the notes; a specific stochastic model was then applied to each context (see Figure 7) [67,68]. Moreover, the maximum entropy model proposed by S. Moulieras and F. Pachet [69] was explicitly based on the assumption that musical expression refers to the local texture, rather than long-range correlations. In this case, the reference repertoires are jazz, pop, and Latin jazz.

Although in-depth studies are necessary before reaching any conclusion, it is our opinion (also based on preliminary informal pop performance analyses) that in the field of popular music, the study of local expressiveness could prove to be of extreme relevance. Most of the research cited in this subsection could find the direct application or be adapted to work in a popular music context.

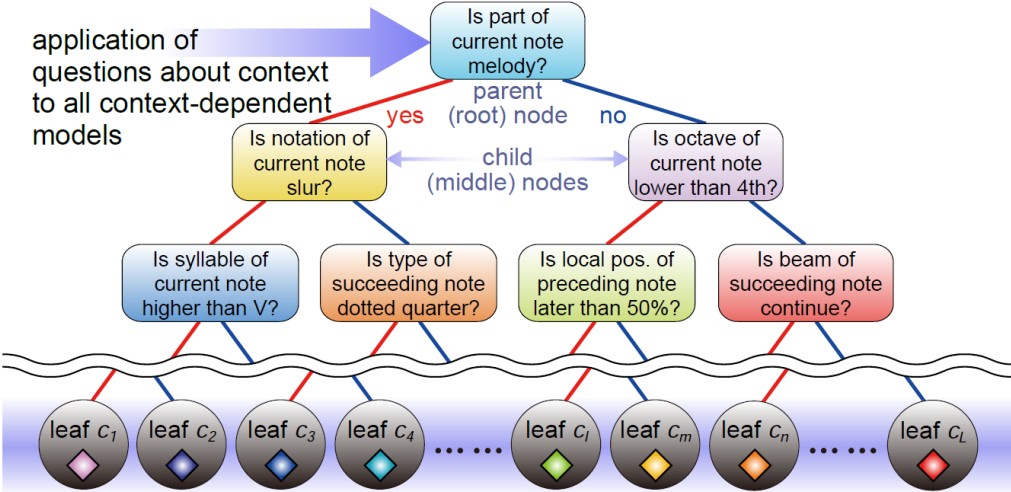

**Figure 7.** Binary tree-based clustering of the local context [68].

*3.5. Score Markings Interpretation*

Peculiarly, M. Grachten and G. Widmer focused (also) on a very specific and limited area of performative musical expressiveness—the interpretation of dynamic score markings [70]. Their model (which must not be considered limited to dynamics alone, but can also be applied to other musical expressive categories) is focused on the explicit expressive markings written in the score (e.g., *crescendo*/*diminuendo* signs, accents, dynamic markings, such as *p*, *f*, *mf*, directly linked to basis functions—numeric descriptors that represent specific components of the score), and investigates how the combination of these markings/*basis functions* influences specific target parameters. In other words, expressive parameters (e.g., note dynamics) are modeled after a weighted linear combination of score data plus noise. In a subsequent work, a Bayesian probabilistic alternative to the original approach to weight estimation (based on least-squares regression) was proposed, together with a new set of *basis functions* and the contextualization of gradual loudness annotations (*crescendo*/*diminuendo*) in relation to the preceding and following notated loudness levels (e.g., $p \rightarrow crescendo \rightarrow f$) [71]. An ANN-based approach (non-linear basis model—NBM) was then investigated [72,73]. In these last contributions, instead of using a simple weighted linear combination of the *basis functions* (LBM—linear basis model), feed-forward neural networks (FFNNs), bidirectional recurrent neural networks (RNNs-see [74]), a combination of FFNN and RNN, and a long short-term memory network (LSTM) were tried on piano solo and symphonic repertoires, showing better prediction accuracy than the original LBM. Another development of the model, therefore, took place with the addition of the relationship between the formation of musical expectations and the corresponding musical performances to the analysis of score features based on basis functions, with significantly positive results [75].

The KTH rule system [47] provides articulation rules, taking into account the markings of legato and staccato present in the score.

While attention to indications present in the score similar to those mentioned above could prove useful in popular music production, the MIDI protocol does not natively provide the tools to describe the score markings, as it is a performance description language more than a symbolic prescriptive language (like the score). In any case, the score features could be included in MIDI in the form of CCs or out-of-range notes, as commonly happens in virtual instruments with multiple articulations. MusicXML or similar approaches may be difficult to integrate into popular music producers' workflows, but innovative ways to harmonize them with MIDI could produce positive results.

*3.6. Relationship between Emotional Intention and Expressive Parameters*

Much of the interest shown by people regarding music is due to the emotional dimension [76]. The research field that investigates music and emotion has grown in recent decades. An excellent general introduction to the topic was offered by P.N. Juslin and P. Laukka [77]. A relatively recent but deep review of the state-of-the-art was presented by T. Eerola and J.K. Vuoskoski [78].

Before delving into more computationally oriented approaches that focus on the relations between music expressiveness and emotion, it seems appropriate to outline the essential characteristics of general reflection on music and emotions.

The first distinction to be made is the one between emotions expressed by the music and identified by the listener, and music-inducted emotions (felt by the listener). According to A. Gabrielsson, there is not necessarily a positive correlation between perceived and felt emotions in music [79]. P. Evans and E. Schubert further investigated the possible relationships between the emotional qualities attributable to musical materials (expressed emotion, called the *external locus of emotion*) and the subjective emotional response to music (felt emotion, called *internal locus*), describing the simple hypothesis of equality between the two as overly simplistic [80]. Moreover, Juslin observed that emotions perceived and expressed in music may be different from each other, and that music does not always arouse an emotional response in the listener [76].

Of fundamental importance in any approach to the description of the relationship between emotion and music is the emotion model adopted. In [78], four models are proposed: discrete model (all emotions can be derived from a limited set of basic emotions, usually fear, anger, disgust, sadness, and happiness); dimensional model (frequently traceable back to J. Russell's circumplex model [81], still relevant today, where emotions can be represented as a mixture of the core dimensions of valence and arousal, in a bi-dimensional space); miscellaneous models (based on a collection of concepts, such as preference, similarity, tension); music-specific models (that focus on emotions that are directly relevant for music, while the other approaches are more general-purpose and may not be fully suited to the music field).

Numerous (configurations of) music features have been linked in past studies to the expressions of discrete emotions (for example, fast tempo, major mode, simple and consonant harmony, and ascending pitch are positively correlated with happiness) [76]. A. Gabrielsson published some of the seminal papers concerning the relationship between parameters, such as timing, dynamics, intonation, and expressed emotion [82,83].

S.R. Livingstone et al. proposed a rule system based on previous studies about music parameters and expressed emotions, capable of modifying not only performance parameters but also score indications, according to the desired emotion [84].

R. Bresin and A. Friberg presented a synthesis approach to the topic: 20 performers were asked to intervene on 7 musical variables (tempo, sound level, articulation, phrasing, register, timbre, and attack speed) simultaneously—and not, as it often happens, one at a time—for communicating 5 different emotional intentions (neutral, happy, scary, peaceful, sad). For each emotion, the mean values and ranges of the musical variables were detected (see Figure 8). These expressive parameters are not dependent on the score to be played [85].

Director Musices is a software available for GNU/Linux, Macintosh, and Windows that implements most of the KTH rule system [86]. The ability of the two to produce virtual performances that can be associated with different emotional states was investigated by R. Bresin and A. Friberg [87].

Several musical cues were analyzed through systematic parameter variations with respect to the emotions expressed by T. Eerola et al.: tempo, register, dynamics, articulation, and timbre. The detected relevance of each cue corresponds to the order in which they are listed above. Another finding is that the musical cues do not seem to have significant interactions; their contribution to the overall expressed emotion appears to be based on simple linear combinations [88].

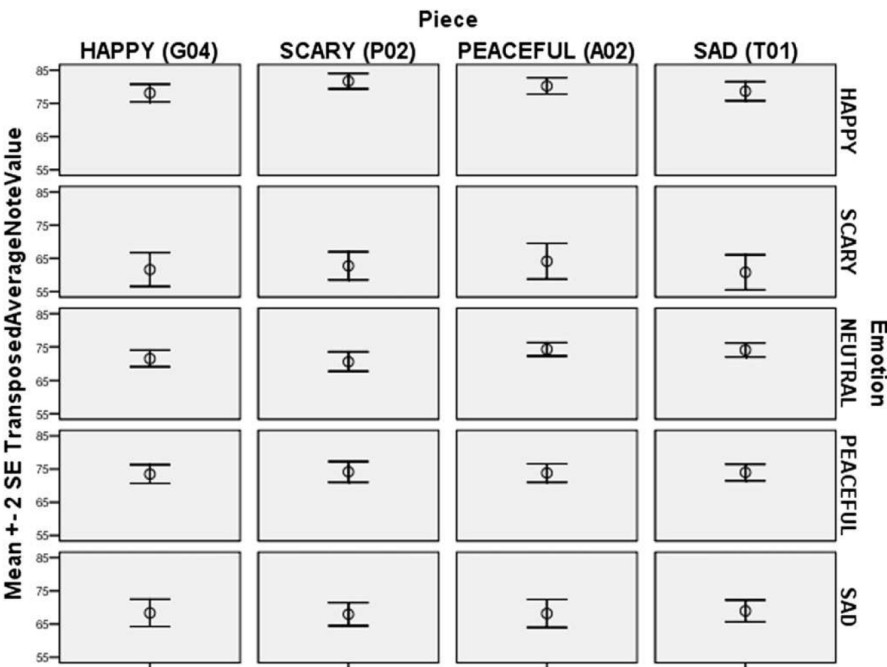

**Figure 8.** Relation between the register and expressed emotion (mean values and range). Reprinted with permission from [85]. 2011, Elsevier.

The origins of these associations may be traced back to analogies with emotional speech [89], human movements [90], and personal and/or cultural associations.

Regarding the induction of emotions, one of the seminal approaches is by L.B. Meyer [50], which bases it on the creation and subsequent confirmation or disruption of expectations, an approach later deepened and extended in E. Narmour's work [51].

P.N. Juslin proposed a unified framework called BRECVEMA, which takes into account eight emotion-induction mechanisms, to be added to the cognitive one: *brain stem reflex* (an emotion is evoked when one or more music parameters exceed a specific threshold), *rhythmic entrainment* (the rhythm of the music influences some internal bodily rhythm of the listener, e.g., the heart rate), *evaluative conditioning* (specific traits of the music are associated with emotions because they were heard many times in specific contexts), *emotional contagion* (the brain responds to specific music stimuli as if they were coming from a human voice that expresses emotion, and mimics that emotion), *visual imagery* (music stimulates imagery of bodily experiences with which it has something in common), *episodic memory* (a connection between the music and personal memories of the listener), *music expectancy* (see the Meyer approach above), and *aesthetic judgment* (the evaluation of the aesthetic value of the piece makes the listener feel emotion) [91].

Most of the research described above could find direct application in popular music-oriented EMP models or could be adapted for this purpose. Popular music seems to be able to evoke and express emotions that are equally powerful to high-art (classical) music, at least when only liked or loved pieces are taken into consideration [92]. Although some research from different perspectives has been conducted on emotion and popular music (see, for example, Y. Song et al.'s research [93]), the commitment is certainly much lower than what has been done in the field of classical music [92]. Much research is needed, but it could definitely be worth it.

### 3.7. Relationship between Sensory Experiences and Expressive Parameters

The expressive intention of a performance can be traced to the desire of expressing sensory perceptions. S. Canazza et al. asked a professional clarinet player to play multiple times the same excerpt taken from W.A. Mozart's *Clarinet Concert in E Major* (K622), once in a scholastic, *normal* way, and the other times trying to express the adjectives *light*, *heavy*,

*soft*, *hard*, *bright*, and *dark*. The recordings were analyzed with regard to time (correlated to amplitude and duration) and frequency (timbre) domains. The data collected were then used to synthesize virtual performances aimed at expressing the above adjectives. A panel of musicians correctly recognized the expressive intention of the computational renditions [94]. In this work, we can already see the bases on which the CaRo system [24,95–97] would later be developed. CaRo will be addressed in detail below, in Section 3.11.

In the pDM system, a real-time sequencer integrated with the KTH rule system—see Section 3.11—a virtual performance can be manipulated through a set of mappers that translates high-level indications into rules parameters. Among them, there are descriptive adjectives, such as hard, light, heavy, or soft [98].

A. Friberg and J. Sundberg compared the stopping of running and the final *ritardando* that marks the termination of a piece of music, noting that they present significant similarities [99].

The association between sensory or physical experiences and performance parameters could be of prime relevance in the development of popular music-oriented EMP systems. Within pop music production teams, these kinds of linguistic parallels often seem to be used [100], could be more easily interpretable by producers, and better convey expressive intentions compared to other terminological families. More in-depth research on the subject, whose surface has only been scratched by the academy, would certainly be valuable to clarify the real scope of this approach to computational expressiveness applied to the popular context.

*3.8. Identification and Modeling of the Performative Styles of Real Musicians*

The study of a specific performer's style can serve multiple purposes: automatic artist recognition, quantitative analysis of the individual style of his/hers, creation of real human-based expressive models.

Visualization and analysis of the performance style of famous artists is one of the explicit objectives mentioned in the work by S. Dixon et al. [28].

Different weight distributions among the rules of the KTH system can be used to represent different performer styles [47]. Director Musices was used to try and reproduce a specific pianist's expressive timing, with good but not optimal results [101]. From the study, it emerges that rule combinations have to change between sections in order to better match the pianist's actual deviations. In the above-mentioned hybridization of KTH rules and ANN [63], after a first step in which the ANN is trained to emulate a selection of the KTH rules, a more complex version of the system, aware of the local context ($n - 1$, $n + 1$, and $n + 2$ note parameters contribute to the definition of the parameters of the current note $n$), is proposed to learn the playing style of a specific pianist. In informal listening tests, better judgment was obtained by the ANN trained with the real pianist, compared to the one trained with the KTH rules.

S.I. Giraldo and R. Ramirez worked on a machine learning system for performative rule-discovering and modeling of expressive jazz guitar performances [102,103]. The ML approach to feature selection, rule discovering, and performance modeling is suitable for research applied to specific individual musicians.

In C. Saunders et al.'s research, tempo and loudness deviations of multiple performers playing the same piece of music were translated into performance worms; from there, general performance alphabets could be derived, and the performances could be represented as strings [104].

Generally speaking, machine learning techniques lend themselves well to the identification of performers. That is the case, for example, in the contributions of E. Stamatatos and G. Widmer [105] and of R. Ramirez et al. [106], respectively, dedicated to the automatic identification of piano and saxophone performers.

The attempt to imitate specific musicians, potentially very different from each other by their nature, imposes an approach to the management of performative parameters that tends to be more agnostic than what occurs in the general modeling of expressiveness in

specific stylistic or historical contexts. Therefore, we see no reason why the research already carried out in the field of recognition and modeling of specific performers should not be applicable in the field of popular music, possibly with some minor adaptations.

### 3.9. Identification of Physical and Psychological Limits of the Performer

While on the one hand, much attention (see previous paragraphs) has been placed on the observations of deviations of the performers attributable to structural high-level or local musical logic, emotional rendering intentions, and sensory references, on the other hand, as L.L. Costalonga et al. observed [107], little research has been devoted to understanding the implications of biomechanical constraints and internal processes of the musician in the implementation of expressive music performance models. L.L. Costalonga et al. called the latter approach *performer-based modeling*; they considered it essential for the development of better EMP systems.

Many studies have been carried out to better understand—among other performance characteristics—motor control, hand dexterity, and timing precision in musicians [108], but it is very rare to see the fallout from these studies on EMP modeling research. The most common objectives in these kinds of studies include the prevention of injuries [109] or the study of optimal fingerings from the biomechanical point of view [110,111].

One excellent exception is the above-mentioned work of Costalonga et al. [107], where biomechanical constraints, errors, noise generation, muscle strength, speed, and endurance are deeply investigated in relation to the guitar, with the declared intention of producing data potentially useful for the development of EMP models.

Another work worthy of mention is that relating to the GERM model, by P. Juslin, A. Friberg, and A. Bresin [1]. The GERM model aims to describe the main sources of variability in expressive performances, placing particular emphasis on the integration of different approaches present in the literature and providing a first unified computational model. GERM is an acronym, which stands for generative rules, emotional expression, random variations, and motion principles. Each area is taken into consideration and dealt with computationally. The generative rules component can be traced back to what is presented in Section 3.3, while the emotional expression component can be placed in the context of what has been addressed in Section 3.6. The last two components, random variations and motion principles, are both relevant to the present subsection. It is observed that stochastic variations are always present in human musical performances, a noise attributable to the limitations of human perceptual and motor skills. These limitations can be associated with motor delay, a noise that does not change depending on the inter-onset interval of the notes (IOI, the time distance between the onsets of two contiguous notes), and with the internal "time-keeper" variance, which changes according to the IOI. Random variations are computed by applying white noise to each note onset time and loudness (motor delay noise), and using filtered white noise (to obtain a *1/f* frequency response, with a variable range depending on the IOI) to simulate the time-keeper noise. Regarding the motion principles, the authors highlight the fact that certain expressive dimensions can be traced back to the imitation of human gestures and of how the human body moves (not only in music-specific contexts). A second kind of biological motion taken into consideration is the one related to the physiological constraints that arise at the intersection between the human body and the movements and gestures required to play a musical instrument. Two rules are implemented in the GERM model, associated with the motion principle area: a ritardando rule based on the observation and modeling of runner decelerations [99], and a phrasing rule based on the phrase arch rule of Director Musices, but taking into account the biological motion issue.

A work that we think could be particularly inspiring in an EMP development perspective (although its declared objectives are not related to that) is the one by P. Visentin et al. concerning the biomechanics of left-hand position changes (shifting) in violin performances [112]. There, the violinist's left-hand position shifts are investigated, with specific regard to EST (end of shift) and DOS (duration of shift) parameters, measured in milliseconds.

Each performer tends to develop a fixed left-hand shifting time (given the space covered by the move is the same), independent of the metronomic tempo and the specific context. This seems to partially contradict B.H. Repp's research [113], in which two pianists are analyzed. The results suggest here that major, cognitively controlled temporal and dynamic parameters of a performance change substantially in proportion to the tempo (unlike what was observed by Visentin et al., according to which the speed of movement remains more or less constant). Anyway, Repp also notes that minor features tend to be determined by tempo-independent motor constraints.

It is our opinion that *performer-based modeling* could play a key role in the future development of EMP models, particularly if they are oriented to attempts at creating computational models of real musicians. This could be particularly true in the popular music EMP field, since from certain points of view the degree of freedom of the performer is more limited there than what happens in western classical music, if only for the fact that, generally, the timing of the performance is bound to a fixed metronomic tempo, or to drums and percussion parts. Some of the previous approaches to expressive timing may not be fully applicable, resulting in the need to follow other strategies to investigate and model performative expressiveness; *performer-based modeling* seems particularly promising in this perspective.

### 3.10. Ensemble Music Modeling

As seen above, most of the computational EMP publications usually deal with solo piano music. Only a few steps regarding expressive ensemble performances have been taken [114]. Among the very few approaches to the topic, we can cite the one by J. Sundberg et al. [115], where an embryonic version of the soon-to-be KTH rule system is proposed, together with a hardware/software system capable of receiving (in input) a music score complemented with phrasing boundaries, harmonies, ties, etc., and returning to output (through the Rulle software) an expressive rendition of the piece. Ensemble synchronization and intonation issues were taken into account. Perfect synchronization between the notes belonging to different instruments but contemporaneous in the score was better judged by a panel of musicians than the competing synchronization solution, based on the freedom of the various instruments, that anyway had to be perfectly in sync once for each bar. In the former approach, all musical parts have to synchronize each time to the most significant one (the one that executes the shortest note) for each note. Concerning the tuning, a general preference for equal temperament over other solutions was detected.

Studies not directly connected to the realization of computational models, but which are of extreme interest in relation to the understanding of the expressiveness-oriented interactions between musicians in popular music, are those by A. Friberg and A. Sundström [116] and M. C. Ellis [117]. They deal with the characteristics of swing rhythm in jazz performances, and they both note that, in jazz ensembles, the instruments do not play in perfect sync, with some instruments slightly lagging behind others. Ellis analyzes three different saxophonists playing over a walking bass line, finding that they all tend to delay the note onsets, relative to the bass, to a variable extent, according to the metronomic tempo (the higher the tempo, the greater the delay). Friberg and Sundström observe how soloists tend to delay the downbeat notes from the drummer's strokes on the ride and play upbeat notes in substantial synchrony with the ride. Moreover, they also describe a correlation between the metronomic tempo and delay of the soloists on the on-beats, but in the opposite direction to what Ellis observed (greater delay on lower tempos).

The modest attention of researchers (to expressiveness in the context of ensemble performances) represents a strong limitation in the field of contemporary popular music production, where more instruments and voices are present at the same time. Any approach to computational expressiveness must be context-sensitive here, and the deviations produced cannot be thought of as if the instrument of interest operated alone.

### 3.11. Conductor Systems

Solutions that involve meaningful user interactions are categorized as conductor systems. In conductor systems, the user can apply, either asynchronously or in real time, changes to the expressive rendering of the musical piece. Two aspects are frequently highlighted in conductor systems research: the capability for users without specific musical training to interact with music, focusing on the creative dimension [97], and the possibility for the user to concentrate on the expressive component of the music, without worrying about the technical difficulties inherent in playing musical instruments [118].

The association between conductor systems and traditional orchestra-conducting inspired some of the earliest research in this area. In M.V. Mathews and J. Lawson work, a radio baton capable of controlling expressive parameters of a synthesizer (timing and dynamics) through the dedicated software Conductor was presented [118–120]. Another example of work involving a virtual orchestra conduction system was that by E. Lee et al.'s [121], based on the recognition of baton gestures and associated time-stretching on the audiovisual recording of a performing orchestra (see Figure 9). A more recent implementation of virtual orchestral conduction was proposed by T. Baba et al. [122].

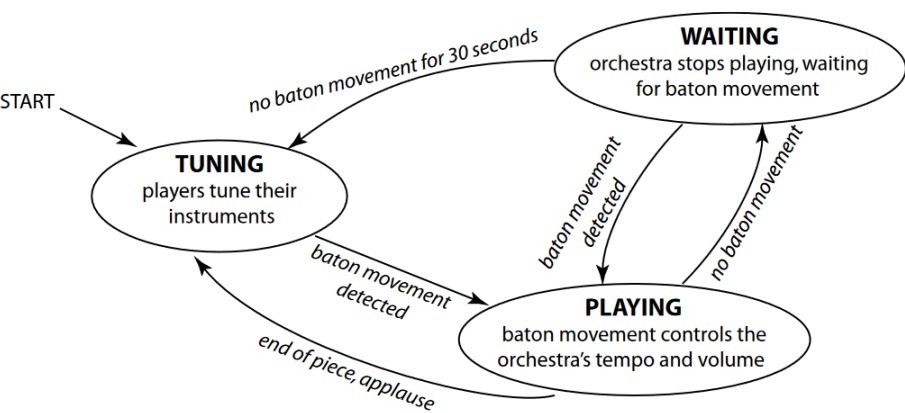

**Figure 9.** State machine for *You're the conductor* [121].

In [123], M.V. Mathews et al. suggested an integration between Conductor software and Director Musices. A real-time extension in pure data language of Director Musices is presented by A. Friberg et al. in [124]. Also correlated to the KTH rule system is the already mentioned work of R. Bresin [63], where the user can interact with the system directly providing input data to the ANN or acting on some of the rules involved in the model.

R. Bresin also proposed a system for the real-time control of pDM parameters [98]. The system provides the visual recognition of human gestures, used to control music parameters at three possible abstraction levels: listener level—the controlling activity is based on basic emotions (happy, sad, angry); simple conductor level—basic overall musical features are controlled using the energy-kinematics space or similar solutions; advanced conductor level—levels 1 and 2 are combined with the explicit control of each beat. In [125], S. Canazza et al. proposed an integration between the KTH rule system, the expressiveness model, and the two-dimensional real-time control space developed at the Centro di Sonologia Computazionale (CSC) [126,127] of the University of Padua (see below).

Of particular relevance is the CaRo system [24,95,96]. It is based on the idea that there are two main sources for musical expressions: the structure of the piece (see Section 3.3) and expressive intention (e.g., bright, dark, hard, see Section 3.7). Asking professional musicians to play the same melody in a neutral, scholastic way, and with a definite set of expressive intentions, it is possible to estimate how performing parameters (e.g., intensity, legato, attack duration, brightness) are affected by the expressive intention of the musician. For each expressive parameter and each expressive intention, two parameters are extracted: k (associated with the mean value) and m (the range of the values, affecting the variance).

The factor loadings obtained from factor analyses (with a two-dimension solution) are then used as coordinates capable of describing the expressive parameters of the performances in a two-dimensional space. Reversing the process, the two-dimensional space can be used to control the expressive intention of neutral performances (see Figure 10). The CaRo system can operate directly on audio signals and MIDI data. Years after the initial release, the CaRo system was adapted to work in a Web 2.0 environment [97].

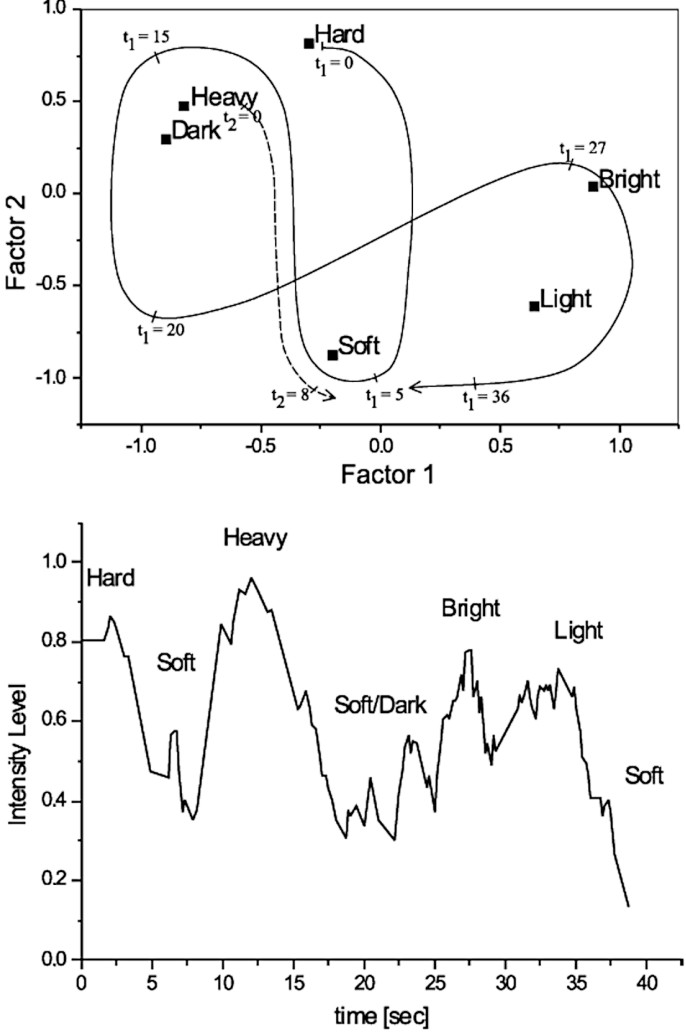

**Figure 10.** The control space movements and the corresponding intensity parameter trend in CaRo. Reprinted with permission from [96]. 2004, IEEE.

In [128], S. Dixon et al. resumed the metaphor of the performance worm [28,29], inverting it, so that tempo and loudness can be controlled in real time by the user. This can be done using hand movements and a digital theremin (air worm), or the computer mouse (mouse worm). In the same contribution, the air tapper and mouse tapper systems are also presented. These can be traced back to the more traditional control of the metronomic tempo, through hand gestures or the use of the mouse.

Generally speaking, conductor systems seem to be of great interest from the point of view of possible applications in the field of popular music production. In popular music, given that the metronomic tempo is generally tied to a steady beat, or to drums and percussion parts, direct control of the same (such as in virtual orchestra conduction solutions) appears to be of little relevance. On the other hand, the control of the loudness parameter, or of the high-level expressive intentions of the performance (sensory or emotional terms

were used in scientific literature, but their appropriateness in the pop world should be verified) should be more relevant.

*3.12. Summary Table*

In the following table, the main features of the most relevant EMP systems reviewed are summarized (in the same order they appear in the paper). In the first cell on the left, after the recall of the research line and the references, a color-coded indication of the thematic areas covered is provided. The correspondences are the following:

- Visual representation: Vis
- Structure-based expressiveness: Str
- Local-based expressiveness: Loc
- Score markings: Sco
- Emotion: Emo
- Sensory experience: Sen
- Identification and modeling of performers: Per
- Physical and psychological limits: Lim
- Ensemble modeling: Ens
- Conductor systems: Con

**Table 1.** Summary table.

| References | Technologies | User Interaction | Main Goal (s) |
|---|---|---|---|
| Performance worm [27–29,128] Vis | MIR through analysis of MIDI or audio data | Tempo and loudness control using hand gestures or PC mouse [128] | Real-time graphical representation of tempo and loudness, user control of tempo and loudness |
| ESP [44] Str Loc | Hierarchical hidden Markov model (HHMM) | / | Expressive performance generation based on the score structure |
| KTH rule system [47,63,87,101,115,124,129] Str Emo Loc Sen Per Con | Rules, ANN [63] | Interaction with the ANN [63], Director Musices and pDM applications [101,124,129] | Expressive music performance generation, modeling of real performers or music performing styles |
| SYVARs [48,59–61] Loc | Statistical research of regularities | / | Find and validate systematic expressive variations in specific contexts |
| Expert system [62] Loc | Analysis by synthesis rule-based expert system | / | Rendition of expressive performances of Bach's fugues |
| Gaussian process regression [65] Loc | Gaussian process ML | / | Expressive performance generation |
| YQX [51] Loc | Bayesian networks | / | Expressive performance generation |
| This time with a feeling [64] Loc | LSTM | / | Generation of solo piano musical parts and their expressive performances at the same time |
| Tonal tension in expressive piano performance [53] Str Loc | RNN LSTM | / | Expressive music performance generation based on the analysis of tonal tensions |

**Table 1.** *Cont.*

| References | Technologies | User Interaction | Main Goal (s) |
|---|---|---|---|
| Laminae [67,68] Loc | Tree-based clustering, Gaussian distributions | / | Expressive performance generation |
| Maximum entropy [69] Loc | Maximum entropy | / | Expressive performance generation, given a specific musical style |
| LBM and NBM [70–73,75] Loc | Linear-weighted combination of parameters [70,71], FFNN RNN LSTM [72,73,75] | / | Modeling of the influence of explicit score markings on expressive parameters. Music expectations considered in [75] |
| Rule system for modifying the score and performance to express emotions [84] Loc | Rules | / | Express emotions modifying not only the performance parameters but also score ones |
| CaRo and CaRo 2.0 [24,95–97] Con Loc Sen | Statistical analysis (principal component) | Real-time interaction through an abstract two-dimensional control space | Graphical description of performances, generation of expressive music performances starting from neutral ones, through user interaction |
| GERM model [1] Str Emo Loc Lim | Rules | / | Expressive performance generation |
| ML approach to jazz guitar solos [102,103,130] Loc Lim | ML ANN, decision trees, SVM, feature selection | / | Discover rules for expressive performance in jazz guitar and expressive model creation through ML techniques |

## 4. Conclusions

In this paper, we briefly present the tools available today to modern popular music producers that can help in building expressive virtual music performances. Although powerful tools for synthesis and sound sampling are commercially available, and there is the possibility of freely varying many of the most relevant expressive parameters, what is missing is the ability to automatically process musical parts provided by the producer—potentially time-quantized and fixed-velocity ones—making them expressive, acting automatically on the tone parameters of the virtual instrument, time, and loudness deviations, articulation, embellishments, or even errors and noise traceable back to the performer's constraints, possibly in a context-aware manner.

In the third Section of the paper, we conducted a reasoned review of the scientific literature dedicated to the EMP sector, evaluated from the point of view of its potential impact in contemporary popular music production. More than one research line could prove useful in helping modern producers.

Graphical representations of expressive parameters can help in the understanding of expression dynamics that may be otherwise hard to catch by a human observer, and of analogies and differences between musical expressiveness in popular music and other contexts. In EMP research, much attention has been paid to the relationship between structure and expressiveness. Part of this research could probably be adapted and applied to popular music, but more investigations are needed before being able to draw any conclusions. Automatic structure identification solutions not based on classical tonality seem to be promising with regard to pop music. The local expressiveness approach is much more relevant in view of the possible applications to pop music. Generally speaking, the data-driven solutions introduced could find applications in pop music. The same can be said about identifying and modeling the performative styles of real musicians.

The emotional intentions and sensory parallelism-related research could find positive applications in popular music, but first of all it would be necessary to better clarify the role of these associations in pop, also from the production practise, psychological, cultural, linguistic and anthropological points of view. Identifying the physical and psychological limits of performers is a topic that has not been particularly well-trodden in EMP research. What has already been done can be of great significance in pop music applications, and more research in the field would be highly desirable. The same can be said about expressive ensemble modeling. Conductor systems are of prime relevance here, because they offer several examples of how it is possible to conceive and implement control systems of performative expressiveness. The score marking interpretation topic could be included in the conductor system as a kind of offline conducting tool.

The main reasons because of which past EMP studies outcomes often cannot be directly applied, or at least not before being rethought and adapted, to the typical workflow of a modern music producer, are:

- The reference repertoire is the classical Euro-cultural one, which presumably may be subject to different rules and practices than popular music. This seems to be confirmed by the fact that similarities and differences between machine learning-induced rules in expressive jazz guitar and rules reported in the literature were found [102];
- Most of the existing EMP literature studies deal with piano. The fact that this is the instrument of choice for research on musical expressiveness can be traced back to technical reasons: being based on the percussion of the strings and not allowing a continuous control of the timbre, as it happens for example in the violin, it is relatively simple to build functional expressive models taking into account a minimum number of parameters (timing and dynamics/velocity) [32,64]. Moreover, hybrid acoustic/digital instruments, such as the Yamaha Disklavier, allow for easy recording of MIDI data from human performances. Furthermore, the piano has a leading role in the western classical repertoire, while in the popular context, its relevance varies according to the genre and artist. It would be useful to extend the instrumental scope of interest to other instruments that are particularly significant in pop music, including drums, electric bass, and guitars;
- The role of music scores is also profoundly different between the classical and popular contexts. While in classical music, the performer offers an interpretation of the composer's intention (given the score written by the latter), in popular music there can be many different situations. Musical parts can be improvised, or there can be only chord charts or lead sheets, which respectively show the harmonic skeleton or the reference melody, which can be embellished and modified even in depth during the performance. If we take into account the declared objective of this contribution, which is to understand which research products could be useful to the contemporary producer in conferring expressiveness to pre-defined parts, and not in the automatic generation of new parts, it is possible to leave out the specific case of free improvisation. As for the relationship and alignment between the real performance and lead sheet, the topic was dealt with in relation to the jazz guitar by S.I. Giraldo and R. Ramirez [103,130];
- In art music, particularly classical music, artists have a lot of freedom to express their individuality [28], while in popular music production, there tend to be more constraints, if only for the fact that often the pieces are recorded with constant metronome tempos, at least in studio productions in recent times (the focus of this contribution). On the other hand, as R. B. Dannenberg and S. Mohan observed [131], in live performances, the metronomic tempo can change both long-term and locally. Dannenberg and Mohan also suggest that statistical models based on the analysis of tempo variations in real performances could be used to generate tempo variations in expressive performance models. This would probably be far from the sensitivity of contemporary pop music producers, but could nonetheless open up new paths in the future;
- In modern popular music, there are usually many instruments and vocal parts played or sung at the same time; most past studies on EMP deal with solo instruments

(notably solo piano), which for obvious reasons have greater freedom of expression, in particular with respect to tempo and timing.

Both academic research and commercial applications, in the reflection or implementation of expressive models or tools, tended (in almost all cases) to separate the generation of control signals from sound synthesis/sampling, dedicating themselves almost exclusively to the control stage (and, therefore, to the generation of MIDI instructions). The aforementioned Synful Orchestra [18] has shown how a more integrated approach is possible (and potentially fruitful). In the academic field, the work of R. B. Dannenberg and I. Derenyi should be noted, who proposed a model of computational expressive performances based on the combination of an instrument model together with a dedicated performance model, capable of generating optimized control signals based on score information [132]. It is a field that has not been explored much, but deserves further study in the future.

In (relatively) recent years, approaches to performative expressiveness based on machine learning techniques have spread. As already noted, these are solutions that, at least at the moment, seem to find the optimal field of application within the scope of local expressiveness. Given the large number of impressive results achieved through ML in almost every field of research, it is understandable that a strong hype about it has grown, but this enthusiasm has not always been accompanied by an adequate reflection on the limits of this kind of approach [133]. Regarding ML in relation to computational expressiveness, we highlight two problematic areas in particular. The first is that the user of expressiveness oriented tools typically needs to control them in a way as transparent as possible, not always a trivial thing to do with ML models. The second involves the need for large quantities of data to train the models, which are not always easy to find, especially when addressing specific musical niches.

In conclusion, it is our hope that, in the future, studies on musical expression will further involve the field of popular music. Part of what has been done in computational EMP can be directly applied or adapted to this specific context, but much remains to be done. This need is justified not only by an expansion of the academic understanding of the phenomenon of musical expressiveness but also by the possible positive effects in the daily work of popular producers and potential commercial applications.

**Author Contributions:** Conceptualization, P.B. and S.C.; methodology, P.B., F.C. and S.C.; investigation, P.B.; writing—original draft preparation, P.B.; writing—review and editing, P.B., F.C., A.R. and S.C.; supervision, S.C. All authors have read and agreed to the published version of the manuscript.

**Funding:** This research received no external funding.

**Conflicts of Interest:** The authors declare no conflict of interest.

## Abbreviations

The following abbreviations are used in this manuscript:

| | |
|---|---|
| ANN | artificial neural network |
| CC | continuous controller/control change |
| DAW | digital audio workstation |
| DOS | duration of shift |
| EMP | expressive music performance |
| EQ | equalization |
| EST | end of shift |
| FFNN | feed forward neural network |
| HHMM | hierarchical hidden Markov model |
| HPF | high-pass filter |
| IOI | inter-onset interval |
| LBM | linear basis model |
| LSTM | long short-term memory |
| MIDI | musical instrument digital interface |

| | |
|---|---|
| MIR | music information retrieval |
| ML | machine learning |
| MPE | midi polyphonic expression |
| NBM | non-linear basis model |
| NIME | new interfaces for musical expression international conference |
| PCM | pulse code modulation |
| RPM | reconstructive phrase modeling |
| RNN | recurrent neural network |
| SVM | support vector machine |
| SYVAR | systematic variations |

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
