# Peer review of "Research in Computational Expressive Music Performance and Popular Music Production: A Potential Field of Application?"

_mti, doi:10.3390/mti7020015_

Round 1
Reviewer 1 Report
This is a very interesting review of the literature along with carefully considered thoughts on how academic research could contribute to an emerging field of expressiveness in popular music production. The paper is clear about key differences between classical music that has been the focus of most academic studies and popular music. I believe this paper does a great job of explaining how a wealth of approaches and idea already exist, yet there are many opportunities to apply and expand this research into the popular music production area.
I have almost no complaints about the paper. Since it is largely a review and speculation, I hope the editors will accept this as a necessary step and contribution in the early stage of an emerging research area.
A few thoughts occurred to me the authors might consider.
First, for section 3.9 or even for the introduction, what about the "GERM" theory, that, if I remember correctly, proposes expressive variations are due to Grammar (music structure), Emotion, Randomness (e.g. neuro-muscular motor noise), and Mechanics, which corresponds to the physical constraints mentioned in Sec. 3.9. I mention this mainly because 3.9 suggests that physical constraints are mostly ignored, whereas I thought the GERM taxonomy was was at least uncontroversial, making physical constraints one of the 4 pillars of expressive deviations.
On page 18, aroun line 707, from the descriptions, it seems that Visentin's findings [105] are completely consistent with Repp's findings [106], yet the text says they are partially contractictory. In what way?
I understand the paper is really about control and not about audio and sound synthesis, but control and sound synthesis are much more interconnected than most realize or acknowledge. This goes back to an engineering approach of "divide and conquer" where it is assumed that music can be decomposed into notes and the note production/synthesis problem reduces to the problem of creating a single note. MIDI and Sampling have further cemented this engineering simplification into our vocabulary to the point that many forget that expressiveness occurs across phrases and cannot necessarily be achieved without attention to sound synthesis. The opposite is true as well: sound synthesis of acoustic instruments that ignores phrasing and timbral variation (i.e. "expression") does not even sound realistic when used to produce phrases, and this is a huge problem for sample-based synthesizers. "In general, the problem with sampling and additive synthesis is the focus on the note rather than either the production of sound or control mechanisms." (Dannenberg and Derenyi, "Combining Instrument and Performance Models for High-Quality Music Synthesis," JNMR 27(3) 1998.) It seems clear that the lines between expressive performance and sound synthesis are fuzzy, and current production/synthesis tools have hardly noticed. Research could contribute a lot here. A good example of one commercial product that came from an academic environment is Lindemann's Synful system which applies fine grain control over note transitions, where context is most important, derived automatically. I think if you listen to examples, you will agree that the results are "expressive" even if the techniques do not exactly match your more limited definition of EMP which to me seems more "outside the note level" than "within and between notes".
There is a growing belief that machine learning will solve all music problems and I think the paper should address the state-of-the-art and issues directly. Some key points might be that (1) commercial music production is almost entirely using human-designed algorithms which have established high expectations for musical and production quality. (2) machine learning suffers from controlability, which is essential in most artistic pursuit, (3) while machine learning for sound synthesis and signal processing is impressive, the absolute audio quality in most cases has been poor and unsuitable for professional use, (4) the need for training data is often at odds with the desire to learn about specific styles or instruments or genres or performers, limiting applicability of ML to real problems. I think some musicians' perspective would be constructive given how many engineers are jumping into the audio processing and MIR fields without much knowledge of studio production.
Another interesting paper you might cite is "Swing Ratios and Ensemble Timing in Jazz Performance: Evidence for a Common Rhythmic Pattern" by Friberg and Sundstrom, and it seems that the control of perceived onset times (vs physical MIDI Note-On time) and the relative timing between different instruments could be important for pop/rock as well as swing.
"Characterizing Tempo Change in Musical Performances" (ICMC 2011, Dannenberg and Mohan) explores tempo variation in popular music. Even if steady tempo is standard in DAWs, there is room for more study of "natural" tempo variation in human performance of popular music and whether it is a defect or a feature.
There are minor English issues, notably p24 line 851: "decline" does not make sense here, and line 853: "conduction" should be "conducting".
Overall, this is a significant contribution to the literature and should be published after some minor revisions.
Author Response
Padua, 20 January 2023
Dear Sirs,
we hereby send our contribution, partially revised in accordance with the indications of the reviewers. First, we want to thank the reviewers, who carried out a thorough critical review and offered valuable indications for the improvement of the article.
In the LaTeX file all changes are tracked, so we will limit ourselves to reporting here the most relevant ones:
• we removed almost all the literal quotations of other contributions, introducing their content into the normal flow of the text, to offer a smoother reading;
• we deepened and offered more references about the nature of authorship in popular music, not attributable to the duality typical of the classical context of composer and performer (lines 36-56);
• we gave space to the work of E. Lindemann (Synful Orchestra), as a significant example of introduction of the search for expressiveness at the level of sound synthesis and not limited to the generation of control signals (lines 188-208);
• we clarified the relevance of NIME, also with respect to contemporary popular production, reporting a significant contribution presented first at NIME, that then became the basis of successful commercial products (lines 220-227);
• we included in the discussion, and therefore in the final summary table, the GERM model, with particular regard to the modeling of physical and psychological limits of the performer (rows 700-726);
• we included in the discussion, in the section on ensemble modeling, the studies on swing rhythm in jazz by A. Friberg/A. Sundström and M. C. Ellis (lines 766-778);
• we reordered the summary table, arranging the entries in the order in which they appear in the paper (for the rest, except for the addition of the GERM model and a few minor adjustments, the table has remained unchanged);
• in the conclusions section we recalled the problem related to the historical focus of attention in the field of research on expressiveness on the dimension of control, to the detriment of that of synthesis (lines 947-957);
• immediately below, we mentioned the limits of the approach to expressiveness with machine learning techniques (lines 958-969).
We hope that we have correctly grasped all the indications and suggestions of the reviewers. In any case, we remain fully available should further action be deemed necessary.
Thanks for your support, best regards,
Pierluigi Bontempi (pierluigi.bontempi@phd.unipd.it)
Sergio Canazza (canazza@dei.unipd.it)
Filippo Carnovalini (filippo.carnovalini@dei.unipd.it)
Antonio Rodà (roda@dei.unipd.it)
Reviewer 2 Report
This paper is ambitious and targets a less-researched area of tooling for music and would be of interest to audiences that are researchers, music industry professionals, and popular music producers at all levels.
While there are improvements that can be made in more succinctly and thoroughly introducing the research and explaining key concepts, this paper becomes particularly interesting in Section 2 and especially 3 when the authors describe and categorise the themes into which EMP approaches can be grouped.
The part on p7-8 re: the literature review by C.E. C-C is a bit inexplicable (see my note in the attached doc).
The visual elements of the paper illustrate the text well and are nice representations of the various themes.
Some of the text could be revised to ensure the sentences are not too convoluted, and the approach to presenting quotes is a bit jarring and unnecessary in my opinion, particularly the OED definitions, but there is also strong, confident and well-synthesised writing.
Overall this paper demonstrates a lot of intelligent work and thinking and is an interesting and useful read.

Author Response
Padua, 20 January 2023
Dear Sirs,
we hereby send our contribution, partially revised in accordance with the indications of the reviewers. First, we want to thank the reviewers, who carried out a thorough critical review and offered valuable indications for the improvement of the article.
In the LaTeX file all changes are tracked, so we will limit ourselves to reporting here the most relevant ones:
- we removed almost all the literal quotations of other contributions, introducing their content into the normal flow of the text, to offer a smoother reading;
- we deepened and offered more references about the nature of authorship in popular music, not attributable to the duality typical of the classical context of composer and performer (lines 36-56);
- we gave space to the work of E. Lindemann (Synful Orchestra), as a significant example of introduction of the search for expressiveness at the level of sound synthesis and not limited to the generation of control signals (lines 188-208);
- we clarified the relevance of NIME, also with respect to contemporary popular production, reporting a significant contribution presented first at NIME, that then became the basis of successful commercial products (lines 220-227);
- we included in the discussion, and therefore in the final summary table, the GERM model, with particular regard to the modeling of physical and psychological limits of the performer (rows 700-726);
- we included in the discussion, in the section on ensemble modeling, the studies on swing rhythm in jazz by A. Friberg/A. Sundström and M. C. Ellis (lines 766-778);
- we reordered the summary table, arranging the entries in the order in which they appear in the paper (for the rest, except for the addition of the GERM model and a few minor adjustments, the table has remained unchanged);
- in the conclusions section we recalled the problem related to the historical focus of attention in the field of research on expressiveness on the dimension of control, to the detriment of that of synthesis (lines 947-957);
- immediately below, we mentioned the limits of the approach to expressiveness with machine learning techniques (lines 958-969).
We hope that we have correctly grasped all the indications and suggestions of the reviewers. In any case, we remain fully available should further action be deemed necessary.
Thanks for your support, best regards,
Pierluigi Bontempi (pierluigi.bontempi@phd.unipd.it)
Sergio Canazza (canazza@dei.unipd.it)
Filippo Carnovalini (filippo.carnovalini@dei.unipd.it)
Antonio Rodà (roda@dei.unipd.it)